# Progress and Perspectives in Designing Flexible Microsupercapacitors

**DOI:** 10.3390/mi12111305

**Published:** 2021-10-24

**Authors:** La Li, Chuqiao Hu, Weijia Liu, Guozhen Shen

**Affiliations:** State Key Laboratory for Superlattices and Microstructures, Institute of Semiconductors, Chinese Academy of Sciences, Beijing 100083, China; lali@semi.ac.cn (L.L.); huchuqiao@semi.ac.cn (C.H.); s20190795@xs.ustb.edu.cn (W.L.)

**Keywords:** flexible, on-chip, energy storage, microsupercapacitor, integrated system

## Abstract

Miniaturized flexible microsupercapacitors (MSCs) that can be integrated into self-powered sensing systems, detecting networks, and implantable devices have shown great potential to perfect the stand-alone functional units owing to the robust security, continuously improved energy density, inherence high power density, and long service life. This review summarizes the recent progress made in the development of flexible MSCs and their application in integrated wearable electronics. To meet requirements for the scalable fabrication, minimization design, and easy integration of the flexible MSC, the typical assembled technologies consist of ink printing, photolithography, screen printing, laser etching, etc., are provided. Then the guidelines regarding the electrochemical performance improvement of the flexible MSC by materials design, devices construction, and electrolyte optimization are considered. The integrated prototypes of flexible MSC-powered systems, such as self-driven photodetection systems, wearable sweat monitoring units are also discussed. Finally, the future challenges and perspectives of flexible MSC are envisioned.

## 1. Introduction

Flexible on-chip microsupercapacitors (MSCs) with advantages of small size, low weight, ease of handling in appearance, ultrahigh power density, and excellent lifespan are of great importance in developing miniaturized, highly integrated, and wearable electronics, where MSC serve the double duty of energy storage and an energy supply unit [1,2,3,4,5,6,7,8,9,10,11]. More specifically, energy harvester devices like nanogenerators that convert energy produced by human motion, walking, mechanical triggering to electrical energy, photoelectrical devices that transform light to electrical energy, and thermoelectrical devices that convert thermal to electrical power storage need an MSC to save the transformed energy [12,13,14,15,16,17,18,19,20]. On the other hand, wearable energy consumption electronics such as sensors, detectors, transistors, etc. require MSC to provide power [21,22,23,24,25,26,27,28,29,30]. As a bridge between energy harvester and consumption devices, the optimization of MSC appears to be particularly urgent [31,32,33].

Since the concept of intelligent wearable electronics was proposed, the on-chip MSC with deformable properties has attracted extensive attentions [34,35,36,37,38]. Up to now, several kinds of MSC devices have been reported, which can be classified with respect to the electrodes, including carbon-based MSC represented by graphene, MXene, carbon nanotube (CNT), transition/metal oxide, transition/metal sulfide, or conducting polymer based MSC, hybrid materials based MSC [39,40,41,42,43,44,45,46,47,48,49,50,51]. Although the emergence of the hybrid materials could improve the electrochemical performance and specific capacitance of the on-chip MSC to a certain extent, the combinations of the two or three materials bring the complex synthesis process, with not all of the combinations showing double or triple performance enhancement [52,53,54]. Therefore, except for seeking the novel electrode materials combination, designing an adaptive method to change the structure of the electrode materials thus realizing the performance improvement of the electrode materials should also be considered [55]. The electrolyte used in fabricating the flexible on-chip is usually in a solid state or an all-solid state, which is well studied along with the development of the electrode materials. Meantime, to realize the practical application of the on-chip MSC, various fabrication technologies have been proposed, such as typical photolithography process, direct laser scribing methods, ink printing procedure, etc.

In this review, we systematically summarize the recent efforts to promote the development of the flexible on-chip MSC and its applications in smart, integrated, and wearable electronics. Targeting the aforementioned problems of the electrochemical performance improvements for a certain electrode material via structural design, we proposed three solutions that contain composites synthesis, 3D architecture build, and in situ modification materials, as shown in Figure 1 [56,57,58,59,60,61,62,63,64,65,66]. In each solution, detailed examples are given to discuss the mechanism of the electro-chemical performance enhancements. In the following parts, the fabrication technologies employed in assembling flexible on-chip MSC are presented, which can be divided into mask, cut, and print methods. Next, based on the MSC in practical application, and the protypes of a smart and integrated system, where MSC served as energy storage in a self-charged system and is used as an energy supply in multifunctional sensing units, are reviewed. At the end of this review, the challenges and future perspectives are proposed for high-performance on-chip MSC and the smart integrated system.

## 2. Electrode Materials Design

In this section, we give a deep discussion of the electrode materials design towards flexible on-chip MSC with high energy density. First, the basic formulas to calculate the specific capacitance, energy density, and power density are provided. Then, composites composed of two or three electrode materials and selection criteria of the various components are listed. Next, a general method of constructing a 3D structure for electrode materials to improve electrochemical performance is introduced. Finally, we will summarize the surface modification to achieve high-performance MSC. The electrochemical performances of the MSC with different electrodes are listed in Table 1.

### 2.1. Calculation Formulas of the On-Chip MSC

The specific capacitance of the on-chip MSC is obtained through cyclic voltammograms (C_v_) based on the following equation:
(1)$$C_{v}=\frac{1}{\mathrm{SVd}}\int_{0}^{V}I\;\mathrm{dV},$$
where C_v_ is the specific capacitance (F/cm^3^); d stands for volume (cm^3^) of finger electrodes, S is the scan rate in cyclic voltammograms (V/s), V represents the potential window (V), and I stands for current (A).

The energy density (E, in Wh/cm^3^) of the on-chip MSC is calculated from the equations:
(2)$$E=\frac{C_{v}{\times V}^{2}}{2\times 3600}\;,$$
where C_v_ is the specific capacitance (F/cm^3^).

The power density (P, in W/cm^3^) of the on-chip MSC is obtained from the equations:
(3)$$P=\frac{E\times 3600}{\Delta t}$$
where Δt represents the total discharge time (in seconds).

### 2.2. Hybrid Electrode Materials

Composite electrode materials have attracted extensive research interest owing to the advantages of the synergistic effect and optimized electrochemical properties [70,71]. In detail, carbon materials with electrical double-layer behavior possess excellent cycling stabilities and robust rate properties, but suffer from the low specific capacitance, while, conducting polymer, metal oxide, or sulfide based pseudo capacitors exhibit high specific capacitance but sustain poor cycling and rate stabilities [72,73,74,75,76,77,78,79,80]. As a result, the composites combined with both the two kinds of electrode materials could exploit their advantages to the full, thus achieving the flexible on-chip MSC with the possibility of practical application [81,82,83,84]. For example, Li et al. proposed a carbon nanotube@polyaniline (PANI) hybrid materials for fabricating on-chip stretchable MSCs, which exhibit a large areal capacitance of 44.13 mF/cm^2^ and offer a high power density of 0.07 mW/cm^2^ at an area energy density of 0.004 mWh/cm^2^ [67]. Chen and co-workers reported a core-shell structural NoMoO_4_@NiS_2_/MoS_2_ nanowire based electrode materials with a high specific capacity of 970 F/g at a current density of 5 A/g, a high energy density of 26.8 Wh/kg at a power density of 700 W/kg [68]. Jung et al. provide a reliable laser-induced ZnO nanorod (NR)/reduced graphene oxide (rGO) -based flexible on-chip MSC, as shown in Figure 2 [56]. Figure 2a,b shows the digital images of the interdigital electrode with ZnO seeds/rGO and ZnO nanorods/rGO materials, respectively. ZnO seed was deposited on rGO film by thermal decomposition of zinc acetate. Then ZnO seed/rGO complexes were placed in the ZnO precursor solution for the hydrothermal growth of ZnO nanorods. Figure 2c,d demonstrates the successfully deposited ZnO seeds and the formation of the ZnO nanorods on the surface of rGO film. The C_V_ curves of the fabricated MSC with different widths of finger electrodes (350, 330, or 310 μm) at a scan rate of 100 mV/s are displayed in Figure 2e. It reveals that 350-ZG MSC has a large average integral area, suggesting a larger stack capacitance of the 350-ZG MSC, which is 3.9 F/cm^3^ based on galvanostatic charge–discharge (GCD) measurements. The Nyquist plots in Figure 2f also suggest the minimal resistance of the 350-ZG MSC. Figure 2g depicts the specific capacitance of the MSC, revealing the high performance of the flexible MSC with wide finger electrodes. All the examples demonstrate the design of composites is an effective strategy for improving the electrochemical performance of the all-solid-state flexible on-chip MSC.

#### 3D Architecture Electrodes

Almost all the electrode materials could transfer their morphology to 1D, 2D, or 3D architectures through electrospinning technology or sacrificial template method without the phase structure change [85,86,87]. Among them, constructing electrode materials into 3D architectures avoids restacking, creates more porousness, and shortens the ion transport distance compared to 1D and 2D materials [55,88]. For instance, Liu et al. used NaCl as a pore-forming agent and glucose as a carbon source to prepare an ultrathin 3D interconnected nitrogen-doped carbon network (N-CN), which then acted as a template to in situ selenylation salinization to synthesize the Co_3_Se_4_@N-CN (CSNC) electrodes [89]. The obtained CSNC electrode materials exhibit excellent lithium storage capacity of 1313.5 mAh/g at the current density of 0.1 A/g, much higher than the pure Co_3_Se_4_ nanoparticles, demonstrating the feasibility of improving the electrochemical performance of certain electrode materials by building 3D blocks. Except for the suggestion of the nanoparticles to 3D interconnected structures, 2D materials that transfer to 3D architectures also have proved enhanced electrochemical performance. Li and co-workers proposed a 3D porous Ti_3_C_2_T*_x_* MXene anode materials and 3D polyaniline@MXene cathode via template method [57]. Figure 3a shows the schematic diagram of the synthesis process of the compressed PANI@M-Ti_3_C_2_T*_x_* electrode. The polystyrene (PS) spheres here were used as a template to make 2D Ti_3_C_2_T*_x_* materials into a 3D open structure, which became a flexible PS@Ti_3_C_2_T*_x_* film by vacuum-assisted filtration. The PS templates were removed after thermal annealing treatment at 450 °C in argon, thus achieving a freestanding and flexible 3D microporous Ti_3_C_2_T*_x_* (3D M-Ti_3_C_2_T*_x_*) anode. PANI@M-Ti_3_C_2_T*_x_* cathode was prepared by a facile drop-and-dry method, as displayed in Figure 3b, the elemental mapping images of the PANI@M-Ti_3_C_2_T*_x_* electrode are well consistent and within the framework of the corresponding SEM images. Figure 3c presents a cross-sectional SEM image of compressed PANI@M-Ti_3_C_2_T*_x_* electrode. The compact PANI@M-Ti_3_C_2_T*_x_* film compressed under 10 MPa can be seen from Figure 3c. Figure 3d provides the C_V_ curves of the PANI@M-Ti_3_C_2_T*_x_* electrode, which delivers an ultrahigh volumetric capacitance of 1632 F/cm^3^ at 10 mV/s and a superior rate capability with 827 F/cm^3^ at 5000 mV/s. Asymmetric SCs were also fabricated with MXene anode and PANI@MXene cathode, which exhibit a high energy density of 50.6 Wh/L and a remarkable power density of 127 kW/L (Figure 3e). Figure 3f shows the large work functions of 1.97 eV for PANI@Ti_3_C_2_(OH)_2_, which is much higher than the 1.61 eV for Ti_3_C_2_(OH)_2_, indicating the higher ability to withstand electron loss and anodic oxidation of the prepared PANI@M-Ti_3_C_2_T*_x_* film.

### 2.3. In Situ Treatment

In situ treatment is also an effective method for improving the electrochemical performance of the electrode materials [90,91,92,93,94]. Our group reported an in situ annealed flexible Ti_3_C_2_T*_x_* cathode based Zn-ion hybrid MSCs with enhanced rate and cycle stability [62]. After annealed treatment, the assembled flexible Zn-ion hybrid MSCs exhibit the maximum areal capacitance of 72.02 mF/cm^2^ (662.53 F/cm^3^) at a scan rate of 10 mV/s and provide a power density of 0.50 mW/cm^2^ at an area energy density of 0.02 mWh/cm^2^. More importantly, the MSC devices present ~80% value of their initial capacitance after 50,000 galvanostatic charge/discharge cycles, which is much higher than MSCs without thermal treatment (54.7%, after 5000 cycles). The ultrastability of the in situ annealed MSCs attribute to the removal of the surface oxygen-containing functional group and the formation of the micropores in Ti_3_C_2_T*_x_* electrode materials. Utilizing in situ treatment of the MSC devices, cycling and rate stability could be improved, and the specific capacitance of the MSC devices also could be increased. For example, Chen and co-workers also proposed an in situ selective surface engineering of GO MSC to improve its specific capacitance [58]. In their work, the rGO based MSC was treated with a pyrrole monomer to achieve selective and spontaneous anchoring of polypyrrole on the microelectrodes without affecting interspaces between the finger electrodes. Figure 4a shows the homogeneous adsorption of pyrrole on the surface of GO filaments owing to the π-π interaction. The oxygen functional groups on GO inducted the self-polymerization of the pyrrole monomer and helped GO reduced to rGO. After this self-oxidation reduction (SOR) reaction, PPy was selectively and accurately anchored on the graphene sheets (pGP), as displayed in Figure 4b. Figure 4c presents the digital images of the fabricated flexible on-chip MSC. The MSC was carbonized, the pGP was changed into NC/rGO. The fabricated MSCs derived from GO, pGP-6 h, pGP-24 h, and pGP-1 week were denoted as MSC-rGO, MSC-6 h, MSC-24 h, and MSC-1 week. The GCD curves in Figure 4d exhibit the capacitances of 13.6, 42.9, 95.3, and 128.4 mF/cm^2^, respectively. The interface-reinforced graphene scaffolds demonstrated a considerably improved specific capacitance from 13.6 to 128.4 mF/cm^2^. Figure 4e depicts the excellent cycling stabilities of the fabricated MSC with capacitance retention of 100% even after 10,000 cycles, demonstrating the self-induced selective interface engineering strategy towards high-performance flexible on-chip MSC.

## 3. Device Fabrication Technology

After preparing the electrode materials, selecting a suitable technology becomes important to realize the large-scale fabrication of the flexible on-chip MSC. In this section, we summarize the manufacturing technology that allows the scalable preparation of flexible on-chip MSC. The mask method is introduced, which means the necessary utilization of a metal mask or lithography mask to fabricate MSC devices. Then cut rote contains mechanical scribing, and laser direct writing method is provided. The ink printing method is presented at the end of this section.

### 3.1. Mask

#### 3.1.1. Photolithography

As a well-developed technology, the photolithography method could prepare metal conductive electrodes with the patterned electrode, high resolution, and minimal size, which is suitable for fabricating on-chip MSC [95]. Figure 5 shows the schematic illustration of the photolithography process for manufacturing the flexible on-chip MSCs with NiFe_2_O_4_ hollow nanotubes electrodes [59]. In a typical procedure, the flexible PET substrate was placed in plasma cleaner for 30 min to enhance the hydrophilicity and wettability. Then, the NiFe_2_O_4_ hollow nanotubes electrodes were dispersed in ethanol and spread on the PET substrate. Subsequently, 35 nm thick Ni film was sputtered on the top of the electrode materials to form the current collector. Then, a conventional photolithographic process was carried out. Next, the superfluous Ni was removed and then treated with air plasma. Finally, after removing the resist, PVA/KOH gel electrolyte was spread on the integrated electrodes of the MSCs. The obtained NiFe_2_O_4_ nanofiber electrodes based on-chip MSC exhibit a specific capacitance of 2.23 F/cm^3^ at the scan rate of 100 mV/s, an energy density of 0.197 mWh/cm^3^, and a power density of 2.07 W/cm^3^. It can be observed that the fabricated MSC following this process always suffers from the low specific capacitance because the active electrode materials were placed under the current collector; therefore, the electrode materials that took part in the actual reaction are very limited. To overcome this problem, we proposed an electrodeposition method to in situ synthesize PPy on the current collect obtained via photolithography [96]. The designed MSC with concentric circles structure shows a large areal capacitance of 47.42 mF/cm^2^ and provides a power density of 0.185 mW/cm^2^ at an area energy density of 0.004 mWh/cm^2^. It’s worth mentioning that the photolithography method offers the possibility of scalable fabricating MSC array, whereas MSC arrays with different series and parallel structures could be manufactured at the same time. Still, the mask needs to be redesigned when we want to change the connection mode; this reason and popular photolithographic techniques involving expensive equipment together with complicated steps significantly increase the cost will prevent the development of photolithography methods in fabricating flexible on-chip MSC.

#### 3.1.2. Screen Printing

The screen printing techniques are simple and cost-effective. Shi et al. reported an ultrahigh-voltage flexible MSC, based on in-series screen-printed rGO on the various substrates [60].

Figure 6a illustrates the schematic diagram of the fabrication process [60]. At first, the conductive ink needs to be prepared by mixing the rGO electrode materials, conducting carbon black and poly(vinyl chloride-co-vinyl acetate) (P-VC/VAc) binder in dimethyl mixed dibasic acid ester (DBE) solvent. The obtained ink exhibits outstanding shear-thinning behavior, allowing for extrusion of the ink through screen meshes under shear force and its quick solidification without shear force (Figure 6b). Figure 6c depicts the photography of the fabricated MSC devices. After that, the various substrates including: flexible PET, A4 paper, glass, or cloth, was put below the customized screen with patterned meshes (mask), the ink was extruded through the screen and deposited on the substrate. After removing the screen and drying the patterned rGO microelectrodes, PVA/H_3_PO_4_ gel electrolyte was dropped on the electrodes of the MSCs, and the all-solid-state rGO based MSC was finally fabricated. Different from the photolithographic technique, screen printing techniques could be scalable for fast and low-cost production and provide complex planar geometries on various substrates. The screen printing method has disadvantages, such as a complex ink preparation process and waste of ink on the surface of the screen, which also leads to the problematic reuse of the screen. The spraying coating method could be seen as the upgraded technique of the screen printing, which doesn’t need to prepare the viscous ink, and the mask could be used repeatedly for many times. Chu et al. demonstrates the large scale fabrication of PANI based MSC array by employing the mask assisted spray-coating method [97]. Gravure printing method is also considered as a promising printing technology owing to its high throughput, optimal control of feature size, and ability to realize large-area manufacturing MSC [98]. For instance, Xiao et al. prepared a MoS_2_@S-rGO based interdigital MSC via gravure printing method, which can be applied in a wide range of electrode materials [99].

### 3.2. Cut

#### 3.2.1. Mechanical Scribing

The mask-free mechanical scribing method has simplified the fabrication process of the MSC devices. Recently, our group reported all-solid-state ZnCo_2_O_4_ nanowires electrode based on-chip MSCs via the mechanical scribing approach [61], as shown in Figure 7. Figure 7a shows the schematic illustration fabricating of flexible all-solid-state on-chip MSCs based on ZnCo_2_O_4_ nanowires electrodes on PET substrate. First, the cleaned PET substrate was put into a plasma cleaner with air flow to enhance hydrophilicity. After that, conductive Ag nanowires were spin-coated on the PET substrate and dried at 60 °C for 10 min. The ink electrode was prepared by mixing ZnCo_2_O_4_ nanowires (75 wt%) and polyvinylidene fluoride (25 wt%) in the proper amount of *N,N*-dimethylformamide solutions. The ink electrode was then spin-coated on the top of the Ag nanowires film and dried at 80 °C for 5 h to remove the remaining organic solution. The mechanical scribing system was designed to fabricate the all-solid-state on-chip MSCs, consisting of a two-dimensional (X-Y axis) moving platform with high-precision guide rails for each axis, a needle mounted vertically over the platform, and a control system. By pre-importing a programmed pattern, on-chip MSCs can be manufactured as the movements of the platform along the X-Y axis. The excess electrode materials were removed by the needle mounted on the platform. Finally, the gel electrolyte of PVA/KOH was dropped on the electrodes to get the final all-solid-state on-chip MSCs. This method allows the large scale fabrication of MSC arrays. Figure 7b displays the digital images of the large scale on-chip 5 × 5 and 10 × 10 MSCs arrays, respectively.

#### 3.2.2. Laser Scribing

The laser direct writing method has demonstrated the universal adaptability, facility, and variable-area patterns with high-resolution, which have no requirement for the solvents, the utilization of binder, additives, and the adjustment of the viscosity, surface tension, and wettability of the materials to be processed [100,101,102,103,104]. Our group reported on a Ti_3_C_2_T*_x_* MXene based Zn-ion hybrid MSCs employing the laser direct writing approach [62]. Figure 8a displays the laser scribing process of the Ti_3_C_2_T*_x_* MXene based Zn-ion hybrid MSCs on the flexible substrate. The spin-coated large-sized Ti_3_C_2_T*_x_* current collector was cut by the laser according to the pre-designed pattern. Then, the Zn anode was prepared via the electrochemical deposition method. Next, the small-sized Ti_3_C_2_T*_x_* cathode was coated on the top of the large-sized Ti_3_C_2_T*_x_* current collector. Finally, PVA/ZnCl_2_ gel electrolyte was spread on the devices. Figure 8b shows the digital photo of the Ti_3_C_2_T*_x_* suspension. The MSC arrays (4 in parallel) could be directly attached to the fingernail, indicating the small-size of the fabricated devices. The digital image of various fine-patterned MXene electrodes, such as “USTB”, “CAS”, “Flextronics”, “Institute of semiconductor” words on a transparent PET substrate (size 3 × 3 cm) is presented in Figure 8c, indicating the universal adaptability of the laser scribing method. A cartoon MSC, and MSC with butterfly-shape, could be quickly and easily fabricated, and demonstrates the possibility to design the MSC based on the wearable electric apparatus. The energy density can be controlled by the series/parallel connections, as shown in Figure 8d. A digital timer driven by the obtained single MSC under bending state and a flexible LED displayer of the “TiC” logo lighted by the MSC arrays under different deformations suggest the great potential application of the MSCs in integrated wearable electronics.

### 3.3. Ink Printing

Ink printing method with printable inks is a promising way for scalable production of flexible on-chip MSC [105,106,107]. Recently, Zhang et al. reported additive-free MXene inks for fabricating MSCs via the extrusion printing and inkjet printing approach [63]. Figure 9a shows the schematic illustration of direct MXene printing using additive-free inks. Figure 9b displays the additive-free MXene inks and the printed MSC devices in series. The additive-free MXene inks possess a good viscosity of ~0.71 Pa s., allowing the direct printing on the untreated plastic and paper substrates with high printing resolution and spatial uniformity. The printed flexible MSCs deliver a high volumetric capacitance up to 562 F/cm^3^ and an energy density of 0.32 µWh/cm^2^, demonstrating great potential in scalable and integrated electronics. Similarly, Liu et al. developed the direct printing method to fabricate exfoliated graphene (EG) based flexible on-chip MSCs [64]. The EG was prepared by the electrochemical exfoliation process in a two-electrode system to expand the graphite foil to EG. The prepared EG ink was dispersed in 2-propanol as EG ink with a concentration of 0.8 mg/mL. To obtain EG based MSC, the EG/PH1000 hybrid ink was used to print the designed pattern on paper or PET substrate. The fabricated devices show a high area capacitance of 1080 μF/cm^2^ at a scan rate of 10 mV/s, and superior rate stability with no obvious capacitance change when the scan rate was increased to 100 mV/s. Besides, ink-jet printing was also performed in work to show the potential of scale-up production. The designed MSC arrays can be easily printed via a “home computer and printer” using the prepared EG ink, as displayed in Figure 9c. Figure 9d demonstrates the lighting test of the 4 MSC devices in-series, opening a new avenue to scalable fabrication of high-performance printable, flexible on-chip MSC.

## 4. Integration and Application

Since the self-charged MSCs by wireless charge circle, photoelectric conversion, and nanogenerator have been insightfully summarized in our previous review, here, we only focus on the novel thermal charged MSC devices in the Section 4.1. In Section 4.2, we provide an MSC powered smart and integrated systems, like MSC powered photo detecting system, multifunctional sensors driven by integrated SCs system.

### 4.1. Self-Charged MSC

All-solid-state flexible on-chip MSC has the advantages of small-size, variable structures, high safety, and comfortable experience, making them one of the best choices for energy supply in highly integrated and low-power wearable electronic devices [108,109]. To meet the new requirements of wearable electronic devices such as long-term independent operation, the energy unit needs to satisfy the self-charging function, thus extending the life of the whole device and broadening the application field. While research has suggested that nanogenerator, solar cells can be integrated with MSC to complete a full energy circulation from acquisition to storage and application, energy acquisition units not only suffer from low efficiency, but also introduce many electronic components in the process of integration, such as AC to DC circuit in generator, which is challenging to realize high safety and comfortable experience [110,111,112,113]. Therefore, it is of great significance to develop all-solid flexible self-charging micro-capacitors. Yu et al. developed a thermally chargeable solid-state SC [114], generating a voltage from a temperature gradient and storing electrical energy in SC like conventional thermoelectrics. Figure 10a shows the working mechanism of thermally chargeable SC. When a temperature gradient is formed between two electrodes, the protons at the hot electrode will migrate to the cold electrode by the Soret effect. Thermodiffusion of protons leads to electrochemical reactions at the two electrodes, when electrons are transferred from the hot side to the cold side by connecting the two electrodes with a load resistor. When the temperature gradient is removed, and the load resistor is disengaged, the protons are randomly distributed [115]. Despite the ion movement, the charges on the bottom electrode remain, completing a charged state of the SC without a temperature difference. Figure 10b depicts the thermally charging behaviors, which can be seen that the SC presents a 0.04 V with ΔT of 5.3 K. The charge-discharge profiles in Figure 10c demonstrate that when a temperature gradient is formed, the SC will spontaneously charge. The thermally chargeable SC generate 38 mV with a large areal capacitance (1200 F/m^2^), paving the way for the future development of self-charged flexible on-chip MSCs.

### 4.2. Integrated System

The development of the Internet of Things (IoT) and big data has demanded more from portable, smart and integrated devices, which require a small-sized energy power unit to help various sensors realize the continuously monitoring of health or the environment. In 2017, our group reported MSCs integrated gas sensing system [69], which contains a Ppy film based circular MSC arrays as energy supply, MWCNT/PANI gas sensor as functionalized units, PCB as signal processing component, Bluetooth as data transmission module, and a phone as an analysis/display terminal, as shown in Figure 11a. The MSC exhibit a volumetric capacitance of 47.42 mF/cm^2^ and a power density of 0.185 mWh/cm^2^. The gas sensor shows a quick response time of 13 s and a recovery time of 4.5 s at room temperature. This wearable system successfully realized the detection and information display of ethanol gas with an unknown concentration (Figure 11b), suggesting its wide application in personalized monitoring drunken driving or detection of ethanol gas in industry. In the following work, we designed a wearable self-powered sweat monitoring system [116]. NiCo_2_O_4_ based MSCs were used to power the NiCo_2_O_4_/chitosan based glucose sensor, ion selective membrane based [Na^+^] and [K^+^] sensors. This smart system can easily and accurately realize the real-time monitoring of perspiration displayed on the individual cellphone to assess personal physiological state by Wi-Fi.

Kim and co-workers reported a body-attached and multisensors integrated system [66], as shown in Figure 11c. The multifunctional integrated system consists of a radio frequency (RF) power receiver, an MSC array, strain sensor, and UV/NO_2_ gas sensor. The MSC could be charged wirelessly, which shows a high volumetric capacitance of 4.7 F/cm^3^, and an energy density of 1.5 mWh/cm^3^ at a power density of 12.6 W/cm^3^. The integrated device was directly attached to the neck of a tester (Figure 11d), indicating its wearability. Figure 11e depicts the motion change curves measured by fragmentized GO foam strain sensor. The GO based strain sensor also could realize the detection of repeated body motion, voice, and swallowing of saliva. The MWNT/SnO_2_ NWs based gas sensor and photodetector (Figure 11f,g) have an excellent response to the NO_2_ gas and UV light. The multifunctional integrated system demonstrated a great potential for practical applications in wearable electronics.

## 5. Conclusions

This review summarizes the recent progress in flexible on-chip MSCs and their application in smart, integrated, and wearable electronics. Various design methods, including composites synthesis, 3D architecture build, and in situ modification materials, have been developed for improving the electrochemical performance of flexible MSC devices. The fabrication technologies used in manufacturing flexible on-chip MSC are presented. Then, for the practical application, the MSC-based integrated system is introduced.

Although considerable progress in flexible on-chip MSC has been achieved, there are also enormous challenges that remain for future practical applications. The existing problems and future directions are as follows:

The electrochemical performance with high energy density still requires to be improved. Many of the composites electrode materials have been developed to fabricate high-performance flexible MSC devices, but the guideline to reveal the selection basis of the electrode materials in composites is rarely available. More attention should be focused on the establishment of a selective standard.Although constructing 3D architecture could improve the electrochemical performance, the pore morphology and size control needs to be considered in further work. The template used to build 3D architecture are expensive and the procedures are complicated. More facile methods and template-free synthesis processes should be developed to get scalable electrode materials with 3D architecture.The exploration of novel technologies may lead to a new achievement in the field of large scale and low-cost fabrication of flexible on-chip MSC. The MSC devices with natures of self-healing, biodegradability, and biocompatibility are expected in the implantable self-powered medical devices and health monitoring devices.For the self-powered systems integrated with functional sensors, the thermal charged MSC could simplify the structure of the integrated system and reduce the energy lost in the energy transformation. Thermal charged on-chip MSC will become an important future direction for direct charging the MSC using the temperature gradient between the human body and the environment.

## Figures and Tables

**Figure 1 micromachines-12-01305-f001:**
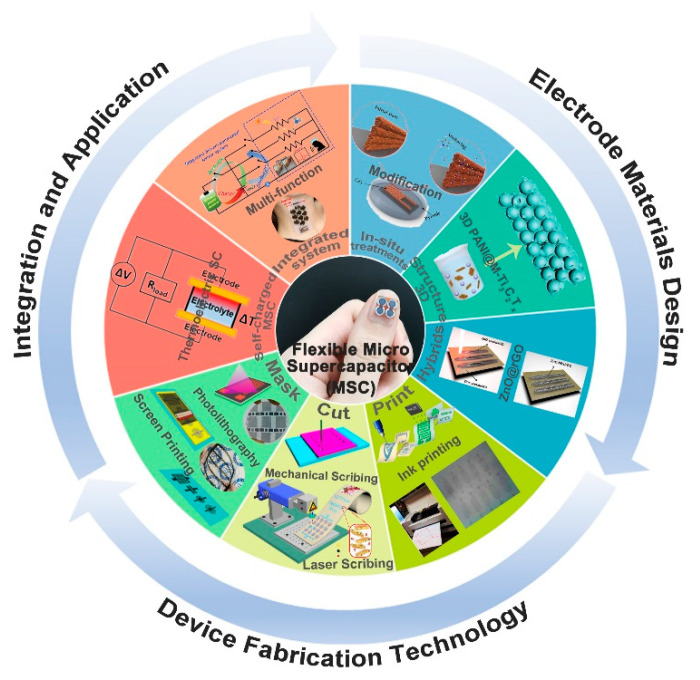
An overview of the flexible on-chip MSCs and their application in smart integrated system. Electrode materials design: Composites—the figure has been reproduced with permission from Springer Nature [56]; 3D architecture—the figure has been reproduced with permission from Wiley [57]; In situ treatment—the figure has been reproduced with permission from Springer Nature [58]; Device fabrication technology: Mask, Photolithography—the figure has been reproduced with permission from The Royal Society of Chemistry [59]; Screen printing—the figure has been reproduced with permission from The Royal Society of Chemistry [60]; Cut, Mechanical scribing—the figure has been reproduced with permission from Wiley [61]; Laser scribing—the figure has been reproduced with permission from Springer Nature [62]; Print, Ink printing—the figure has been reproduced with permission from Springer Nature [63] and Wiley [64]; Integration and application: Self-charged MSC—the figure has been reproduced with permission from Wiley [65]; Integrated system—the figure has been reproduced with permission from Wiley [66].

**Figure 2 micromachines-12-01305-f002:**
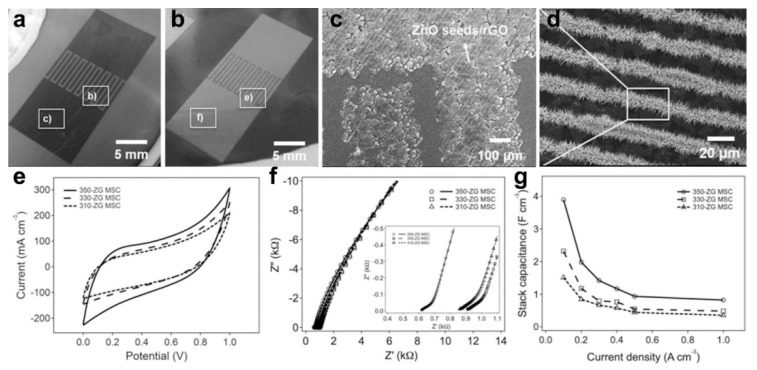
ZnO/rGO composites based on-chip MSC. (**a**,**b**) Photograph of the interdigital electrode with ZnO seeds/rGO and ZnO nanorods/rGO materials, respectively; (**c**) SEM image of the ZnO/rGO finger electrode; (**d**) High-solution SEM image of the ZnO/rGO electrodes; (**e**) C_V_ curves of the ZnO/rGO based on-chip MSC; (**f**) Nyquist plots; (**g**) Specific capacitance of the fabricated MSC. The figure has been reproduced with permission from Springer Nature [56].

**Figure 3 micromachines-12-01305-f003:**
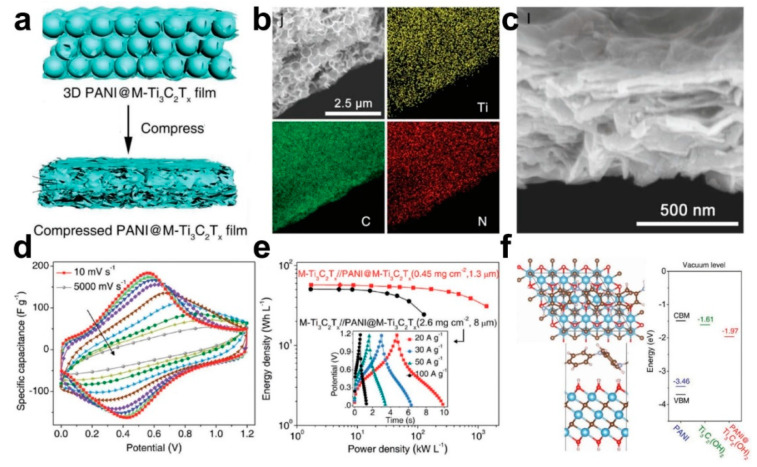
PANI@M-Ti_3_C_2_T*_x_* electrode materials with 3D structure. (**a**) Schematic diagram showing the synthesis process of the compressed PANI@M-Ti_3_C_2_T*_x_* electrode; (**b**) SEM and corresponding elemental mapping images of PANI@M-Ti_3_C_2_T*_x_* electrode; (**c**) Cross-sectional SEM image of compressed PANI@M-Ti_3_C_2_T*_x_* electrode; (**d**) CV curves of the asymmetric M-Ti_3_C_2_T*_x_*//PANI@M-Ti_3_C_2_T*_x_* SC; (**e**) Volumetric energy and power densities; (**f**) Work functions of the electrodes. The figure has been reproduced with permission from Wiley [57].

**Figure 4 micromachines-12-01305-f004:**
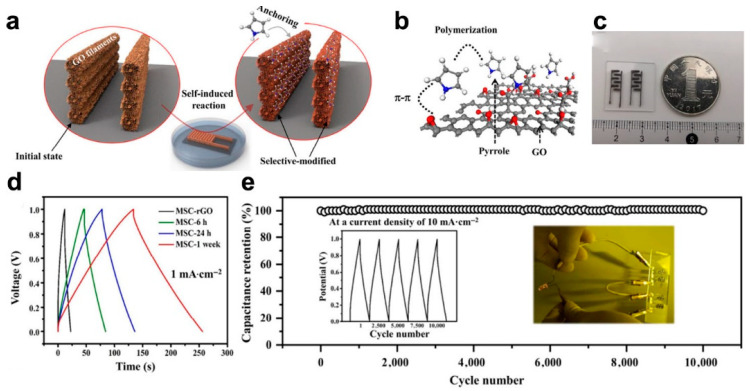
In situ modified rGO with polymer. (**a**) Schematic illustration for growth pyrrole on the surface of rGO micro-patterned electrode; (**b**) The interaction between rGO and pyrrole; (**c**) Photograph of the printed rGo/Ppy based MSC; (**d**) GCD profiles of the MSC; (**e**) Cycling stability of the MSC. The figure has been reproduced with permission from Springer Nature [58].

**Figure 5 micromachines-12-01305-f005:**
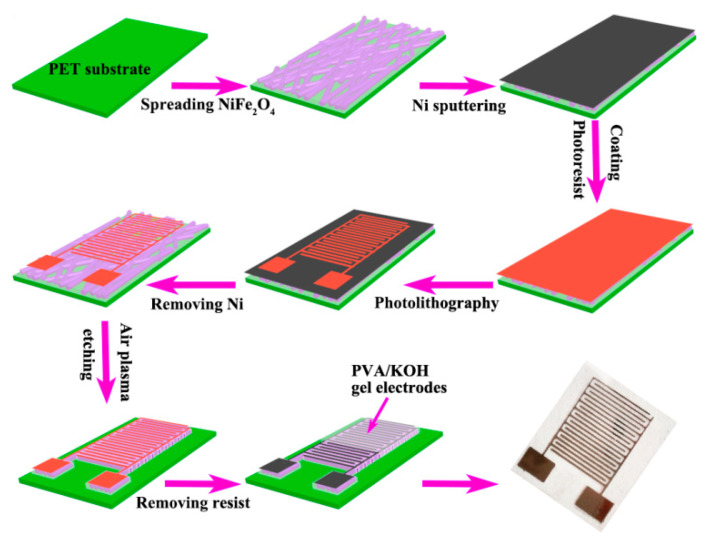
Schematic illustration of the photolithography process for fabricating the flexible on-chip MSCs with NiFe_2_O_4_ hollow nanotubes electrodes. The figure has been reproduced with permission from The Royal Society of Chemistry [59].

**Figure 6 micromachines-12-01305-f006:**
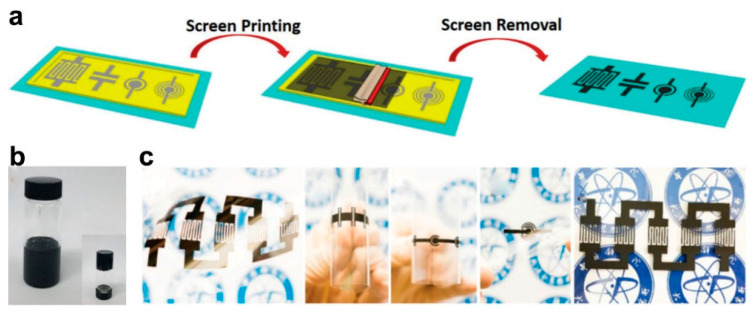
(**a**) Schematic of the screen printing process for fabricating the rGO based flexible on-chip MSCs; (**b**) Digital image of the prepared rGO ink; (**c**) Photography of the fabricated MSC devices. The figure has been reproduced with permission from The Royal Society of Chemistry [60].

**Figure 7 micromachines-12-01305-f007:**
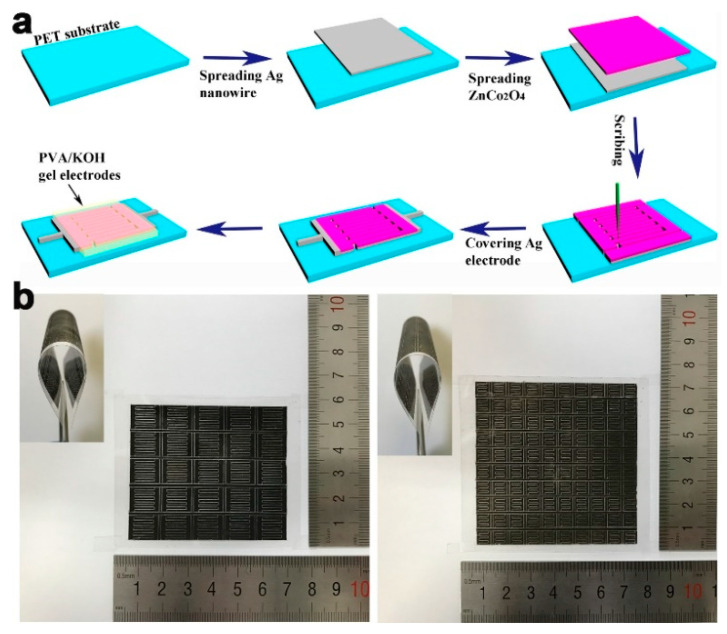
(**a**) Schematic illustration for the fabrication of flexible all-solid-state on-chip MSCs based on ZnCo_2_O_4_ nanowires electrodes on PET substrate; (**b**) Photographs of the large scale on-chip MSCs arrays. The figure has been reproduced with permission from Wiley [61].

**Figure 8 micromachines-12-01305-f008:**
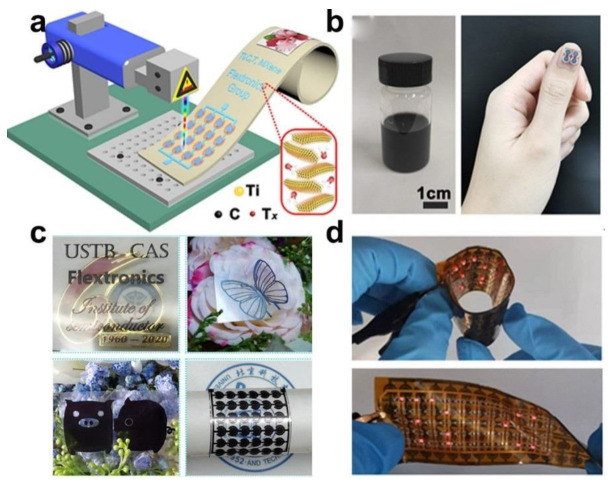
Schematic illustration of the laser direct writing process for fabricating the flexible on-chip MSCs. (**a**) The schematic diagram exhibiting the fabrication process; (**b**) Digital photos of the Ti_3_C_2_T*_x_* supernation and the fabricated micro device array directly attached to the fingernail; (**c**) Optical images of the laser written MSC array; (**d**) Digital images of the Ti_3_C_2_T*_x_* based Zn-ion MSC array powering a flexible LED array of the “TiC” logo under different deformations. The figure has been reproduced with permission from Springer Nature [62].

**Figure 9 micromachines-12-01305-f009:**
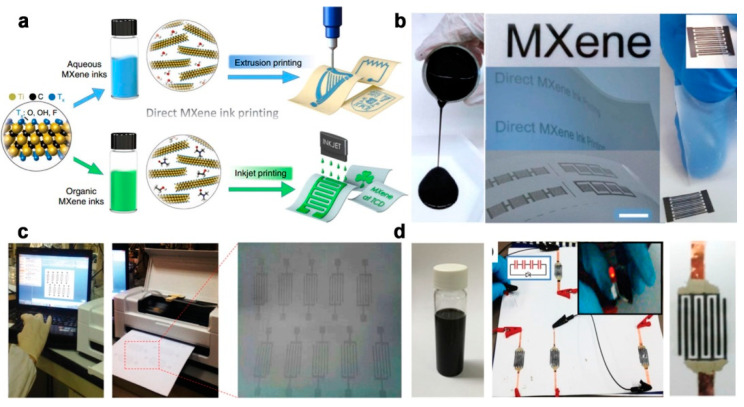
(**a**) Schematic illustration of the ink printing process for fabricating the flexible on-chip MSCs; (**b**) Photography of the MXene ink and printed MSC devices. The figure has been reproduced with permission from Springer Nature [63]; (**c**) Inkjet printing of MSC array via a “home computer and printer” using prepared EG ink; (**d**) Photography of the EG ink and printed MSC devices. The figure has been reproduced with permission from Wiley [64].

**Figure 10 micromachines-12-01305-f010:**
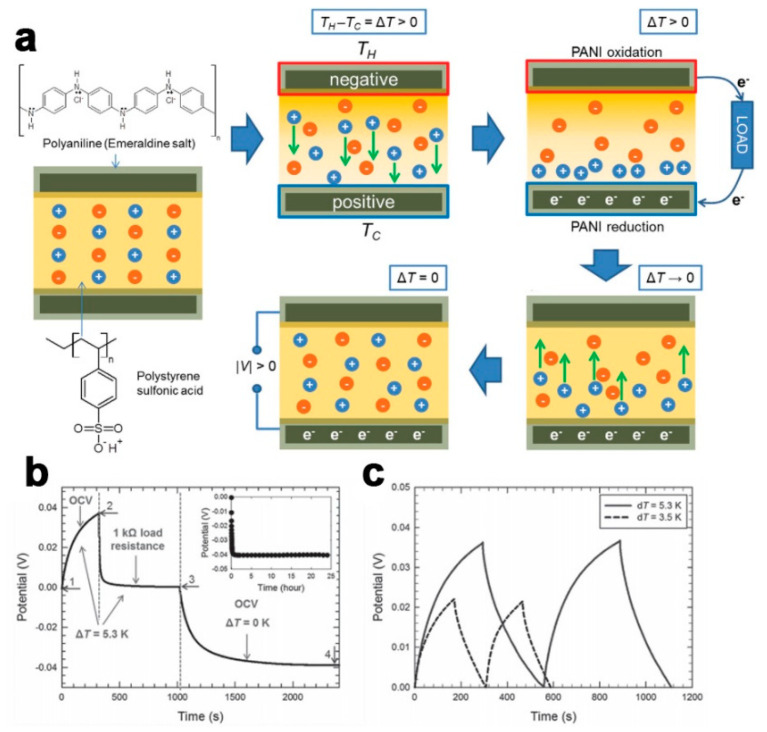
(**a**) Working mechanism of thermally chargeable SC; (**b**) Thermally charging behaviors with ΔT of 5.3 K; (**c**) Thermally charging SC and discharging at a constant current. The figure has been reproduced with permission from Wiley [113].

**Figure 11 micromachines-12-01305-f011:**
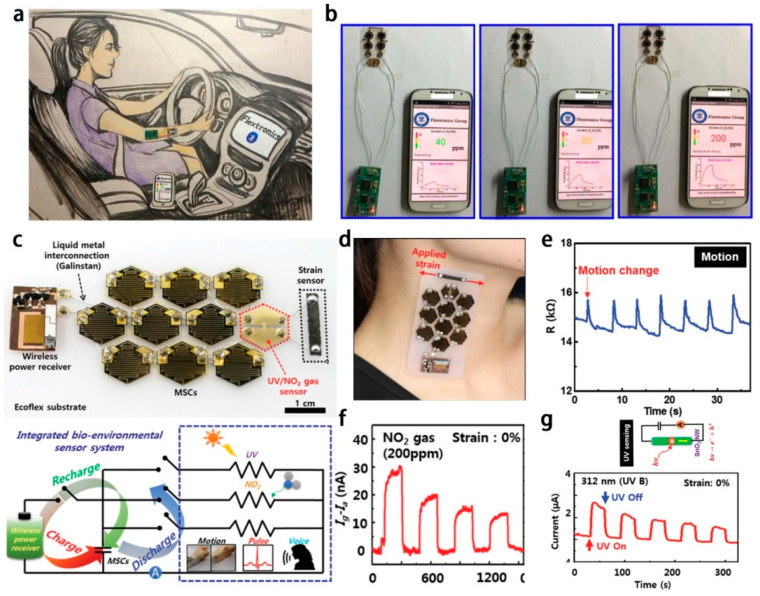
(**a**) Picture of a subject wearing the individual MSC array-gas sensor-analysis unit; (**b**) Real-time C_2_H_5_OH concentration analysis/display; The figure has been reproduced with permission from Elsevier [69]; (**c**) Photography and circuit diagram of the multifunctional integrated system; (**d**) Digital image of the prepared integrated devices; (**e**) The pressure sensor behavior; (**f**) The gas sensor behavior; (**g**) The photo response of the devices. The figure has been reproduced with permission from Wiley [66].

**Table 1 micromachines-12-01305-t001:** Summary of flexible MSCs with different electrode materials and their electrochemical performances.

Electrodes	Specific Capacitance	Energy Density	Power Density	Ref.
PANI//Zn	250 µAh/cm^2^	0.25 mWh/cm^2^	0.99 mW/cm^2^	[2]
Active carbon	0.32 mF/cm^2^	0.3 μWh/cm^2^	66.5 μW/cm^2^	[3]
rGO/PEDOT	7.7 F/cm^3^ at 0.02 A/cm^3^	5 mWh/cm^3^	141 W/cm^3^	[4]
Co_3_O_4_/Pt	35.7 F/cm^3^ at 20 mV/s	3.17 mWh/cm^3^	47.4 W/cm^3^	[5]
rGO/MWCNT	49.35 F/cm^3^ at 20 mA/cm^3^	47 mWh/cm^3^	10 mW/cm^3^	[6]
rGO fiber	121 F/cm^3^ below 1 V/s	0.01 Wh/cm^3^	100 W/cm^3^	[7]
rGO	10.38 mF/cm^2^	1.08 mWh/cm^3^	83.5 mW/cm^3^	[32]
MoO_3-x_ nanorod	41.7 mF/cm^2^	5.8 μWh/cm^2^	-	[35]
carbon/Cu nanowire	7.43 F/cm^3^ at 0.17 mA/cm^2^	0.66 mWh/cm^3^	0.36 W/cm^3^	[41]
ZnO/rGO	3.9 F/cm^3^	0.43 mWh/cm^3^	0.13 mWh/cm^3^	[56]
Ti_3_C_2_T*_x_* MXene/ PANI@MXene	1632 F/cm^3^ at 10 mV/s	50.6 Wh/L	127 kW/L	[57]
NiFe_2_O_4_	2.23 F/cm^3^ at 100 mV/s	0.197 mWh/cm^3^	2.07 W/cm^3^	[59]
rGO	1.0 mF/cm^3^ at 5 mV/s	1.81 mWh/cm^3^	297 mW/cm^3^	[60]
ZnCo_2_O_4_		0.065 μWh/cm^2^	0.092 mW/cm^2^	[61]
Ti_3_C_2_T*_x_*//Zn	662.53 F/cm^3^	0.02 mWh/cm^2^	0.50 mW/cm^2^	[62]
Ti_3_C_2_T*_x_*	562 F/cm^3^	0.32 µWh/cm^2^	11.4 µW/cm^2^	[63]
rGO	2 mF/cm^2^ at 5 mV/s	-	-	[64]
CNT/PANI	44.13 mF/cm^2^	0.004 mWh/cm^2^	0.07 mW/cm^2^	[67]
NoMoO_4_@NiS_2_/MoS_2_	970 F/g	26.8 Wh/kg	700 W/kg	[68]
Ppy	47.42 mF/cm^2^	0.004 mWh/cm^2^	0.185 mW/cm^2^	[69]

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
