# Peer review of "Progress and Perspectives in Designing Flexible Microsupercapacitors"

_micromachines, 2021, doi:10.3390/mi12111305_

Round 1

Reviewer 1 Report

This manuscript gives an updated summary of recent development of designing flexible microsupercapacitors and can be accepted after minor revision.

1# In Device Fabrication Technology, in addition to screen printing and inkjet printing, other printing methods, such as gravure printing, should also be introduced, as gravure printing is a roll-to-roll process for large-area manufacturing.

2# In 3.2.2 Laser Writing, it is better for authors to change to “laser scribing”. In my opinion, laser writing contains cutting. But the example (Ref. 58) authors provide is more likely “laser scribing”.

3# In 4.2 or conclusions, I think the potential of MSC for internet of things or big data should be summarized. I think it should be a hot topic.

Reviewer 2 Report

The review “Progress and Perspectives in Designing Flexible Microsupercapacitors” summarized the recent progress made in the development of flexible MSCs and their application in integrated wearable electronics. The topic is interesting and the manuscript is generally well-written. But before publish on this journal, several comments should be addressed.

  1. Figure 1. The words in this figure are not clear. Please enlarge the words or remove them.
  2. There are too many grammar mistakes in the manuscript. For example, Page 1 “on the other end”; Page 1, “Although the emergence of the hybrid materials could improve the electrochemical performance and specific capacitance of the onchip MSC to a certain extent, the combination of the two or three materials brings the complex synthesis process, and not all the combination shows a double or triple performance enhancement” Page 2 “summarize”; Page 2, “we give a deep discussion”… Please revise the whole manuscript carefully.
  3. It is strange to put the section “composites” in 2.2. It is better to change the section title.
  4. 3.3. D architecture. Please double check the section title.
  5. Please double check the logical connection in the section 2 and section 3 between each part.

Reviewer 3 Report

In the current manuscript, the authors reviewed the recent progress of flexible micro-supercapacitors. The manuscript is well written. A brief overview on the progress of flexible microsupercapacitor has been well-demonstrated. Therefore, I recommend acceptance of this manuscript after minor revision.    

  1. Few recently published articles on supercapacitors/energy storage devices like 1. Nano Research 12 (11), 2655-2694, 2019; 2. Progress in Energy and Combustion Science 75, 100786, 2019; 3. Materials Today, 39, 47-65 2020, 4.RSC Adv., 2016,6, 84769-84776, 5. Sustainable Energy Fuels, 2021,5, 1235-1254, 6.  Nat Commun12, 2647 (2021). https://doi.org/10.1038/s41467-021-22912-8, 7. https://doi.org/10.1002/ente.202000844 should be cited in the introduction part of the manuscript.
  2. In the current manuscript, although the authors discussed more about the technology, the types of electrode materials have not been discussed in details. Therefore, the authors are requested to discuss about the electrode materials in a separate section.
  3. The major advantages of microsupercapacitors should be discussed in details in the introduction part.
  4. A separate table on the electrochemical performance of different types of microsupercapacitors should be included.
